# Pilot Study to Develop and Test Palliative Care Quality Indicators for Nursing Homes

**DOI:** 10.3390/ijerph18020829

**Published:** 2021-01-19

**Authors:** Charlèss Dupont, Robrecht De Schreye, Joachim Cohen, Mark De Ridder, Lieve Van den Block, Luc Deliens, Kathleen Leemans

**Affiliations:** 1VUB-UGhent End-of-life Care Research Group, Vrije Universiteit Brussel (VUB), 1090 Brussel, Belgium; robrecht.deschreye@sciensano.be (R.D.S.); Joachim.Cohen@vub.be (J.C.); lieve.van.den.block@vub.be (L.V.d.B.); Luc.Deliens@vub.be (L.D.); 2Department of Family Medicine and Chronic Care, Vrije Universiteit Brussel (VUB), 1090 Brussel, Belgium; 3Department of Radiotherapy, University Hospital Brussels, 1090 Brussel, Belgium; Mark.DeRidder@uzbrussel.be; 4Department of Public Health and Primary Care, Ghent University, 9000 Ghent, Belgium

**Keywords:** nursing homes, quality indicators, quality measurement, palliative care, quality of care, end of life care

## Abstract

An increasingly frail population in nursing homes accentuates the need for high quality care at the end of life and better access to palliative care in this context. Implementation of palliative care and its outcomes can be monitored by using quality indicators. Therefore, we developed a quality indicator set for palliative care in nursing homes and a tailored measurement procedure while using a mixed-methods design. We developed the instrument in three phases: (1) literature search, (2) interviews with experts, and (3) indicator and measurement selection by expert consensus (RAND/UCLA). Second, we pilot tested and evaluated the instrument in nine nursing homes in Flanders, Belgium. After identifying 26 indicators in the literature and expert interviews, 19 of them were selected through expert consensus. Setting-specific themes were advance care planning, autonomy, and communication with family. The quantitative and qualitative analyses showed that the indicators were measurable, had good preliminary face validity and discriminative power, and were considered to be useful in terms of quality monitoring according to the caregivers. The quality indicators can be used in a large implementation study and process evaluation in order to achieve continuous monitoring of the access to palliative care for all of the residents in nursing homes.

## 1. Background

In the past decade, in many Western countries, an increasing number of elderly persons were admitted to nursing homes. Projection studies concerning numbers of deaths and place of death suggest that, by the year 2040, the majority of deaths will occur in nursing homes [1,2]. Moreover, prediction studies also indicate that the need for high quality care at the end of life will most likely double in the nursing home setting, because of an increasing prevalence of frailty and multimorbidity in the resident population [2,3], being linked to a strongly reduced average length of stay of residents in recent years [4]. The quality of care at the end of life in long-term care facilities is currently high on the agenda: WHO and other health care organizations have advocated for good palliative care for older people already for years [5,6]. Additionally, in research, emphasis in the past decade has been placed on the quality of advance care planning, autonomy of residents, and the implementation of palliative care in nursing homes [7,8]. Until now, palliative care has been insufficiently developed in nursing homes and international studies show the late initiation of palliative care and even mostly for residents with a cancer diagnosis. An urgent need arises for better access for all elderly persons to high quality palliative care provision in the nursing home context.

When implementing palliative care in a nursing home context, it is important to evaluate its success and measure its outcomes for residents. One way to do so is the use of quality indicators within a continuous cycle of implementing, monitoring, and improvement [9]. By continuously monitoring quality of care and its outcomes and conducting implementation trajectories that are based on the results of these measurements, care teams are able to optimize the quality of care based on information, patient experiences, and best practice examples [10,11,12,13]. Quality indicators can be used within this monitoring cycle to provide data on subjective and objective aspects of quality of care over time. They are defined as measurable aspects of care, calculated as a percentage with a predefined numerator and denominator [14,15]. These indicators give caregivers information on their performance in terms of care processes and outcomes and which elements of care may need improvement [16]. Several national health care monitoring programs have been started in Western Countries, including Belgium, based on quality indicators. However, they mainly focus on the hospital or home setting or circumstances surrounding death such as symptoms and place of death [17,18,19,20,21,22,23,24,25]. Although initiatives have been taken for improving palliative care in nursing homes [24,26,27], researchers were not yet able to validate and implement solid quality indicators for palliative care in this specific setting. 

A previous program to develop and implement palliative care quality indicators, the Belgian Q-PAC study, used a rigorous development method combining literature review, expert consultation, and pilot testing, resulting in a core set of 31 quality indicators covering a broad range of aspects of palliative care. The set was meant to be used for all palliative care services and settings, including nursing homes [24,28,29]. However, because the nursing home setting appeared to be too different from the specialized palliative care services in terms of organization and structure of care (e.g., no dedicated palliative care teams), and in characteristics of the population cared for (e.g., specific population with frailty, dementia, cognitive decline), the quality indicator set was implemented into all specialized palliative care services, but not in the nursing homes from 2014 onwards [30]. Therefore, a need persisted to investigate which indicators can be used for monitoring the quality of palliative care in the nursing home context. 

Because of the increasing need for development of palliative care and its monitoring in nursing homes, we started a project to develop a set of quality indicators for the quality of palliative care in nursing homes. Previous research already highlighted the importance of person centered care through autonomy and involvement of family, but also communication and advance care planning in nursing homes [31,32,33,34]; hence, we decided to develop quality indicators specifically targeting advance care planning, palliative care, and end of life care. The main aim for nursing home teams is to obtain insights in their care processes and outcomes, and further develop missing elements in the care for their residents. In this study, we develop and test a quality indicator set and measurement procedure for palliative care in nursing home context. 

## 2. Methods 

### 2.1. Design

We used a two-step approach with a mixed-method design based on a standardized indicator testing protocol for generic quality indicators in order to develop the quality indicator set and measurement procedure [29,35]. First, we developed the set of quality indicators using the Rand/UCLA appropriateness method in three phases: (1) literature review to develop a preliminary set of quality indicators, (2) interviews with experts to test face validity of the preliminary set of quality indicators, and (3) indicator selection by expert evaluation [36]. Second, we evaluated the face validity, feasibility, discriminative power, and usefulness of the quality indicators in a quantitative cross-sectional application of the quality indicators in combination with qualitative interviews. Figure 1 shows an overview of the development process and pilot testing.

### 2.2. Step 1: Indicator and Questionnaire Development 

#### 2.2.1. Phase 1 and 2: Literature Study and Expert Interviews

In phase 1, we identified a comprehensive set of candidate quality indicators for palliative care in nursing homes. Therefore, we adapted the existing Belgium quality indicators for specialized palliative care listed in the Q-PAC (Quality indicators for Palliative Care) study [24] to the nursing home context. To do so, we searched literature in PubMed while using a snowball method starting with the reviews of Pasman et al. and De Roo et al. [37,38]. We searched for existing quality indicators, domains of quality of care for elderly and questionnaires or instruments for quality of palliative care, advance care planning, and end-of-life care in the nursing home context. Candidate quality indicators could be processed or outcome indicators and emphasis were placed particularly on subjective quality indicators in order to reflect the user perspective on quality of care. In phase 2, we performed interviews with relevant stakeholders (i.e., healthcare professionals, community-based organizations, and policy makers) that are involved in Flemish nursing home care, to test face the validity of the candidate quality indicators and gather additional indicators and domains for this specific context not found in literature. This way, an iterative process of literature search and interviews lead to the selection of a preliminary set of quality indicators representing all of the identified domains. Furthermore, to be able to calculate the quality indicators, we operationalized each of them into questions for residents, bereaved family and nursing home staff, accompanied with measurement instructions. This was done based on input that was provided by the experts and questionnaires identified in the literature.

#### 2.2.2. Phase 3: Expert Consensus 

The preliminary set of candidate quality indicators was sent to 15 experts (see results infra). They were asked to score the quality indicators with “1” as “not appropriate” to “9” as “very appropriate” in order to measure the quality of palliative care in nursing homes. Experts were provided with the candidate indicator’s description, rationale, numerator and denominator, question (per response type: residents, bereaved family, or nursing home staff) and literature source. They were also able to suggest missing domains or themes.

The median scores on appropriateness were calculated per candidate indicator. The quality indicators were then categorized based on the RAND/UCLA consensus method: accepted, to be decided, or rejected. Indicators were immediately accepted if they had a median score of 7 or more and if no more than two experts scored the indicator with a 1, 2, or 3 (strict positive consensus). Indicators with a median score of 3 or lower and for which no more than two experts scored the indicator 7, 8, or 9 were immediately rejected (strict negative consensus) [36]. All other to-be-decided indicators were discussed during the one-day plenary discussion until consensus was found regarding rejection or acceptance among the experts. During the discussion, additional selection criteria were to be considered as defined by the researchers: (1) a maximum of eight quality indicators per questionnaire (i.e., response type) was suggested to ensure feasibility of the quality monitoring in nursing homes, without overburdening nursing home staff, and (2) experts were encouraged to consider a good balance between process and outcome indicators, as well as objective and subjective quality indicators. As such, experts were asked to select the eight most important indicators for each response type. 

#### 2.2.3. Questionnaires to Measure the Quality Indicators 

After defining the quality indicators together with the experts, we developed four questionnaires to be able to calculate the performance score per quality indicator. These questionnaires were based on questions of validated scales as much as possible, or if no good question gathering the right information for a specific indicator existed in the literature, it was developed by the researchers, together with the experts. An overview of all indicators, accompanying questions and evidence can be found in the Appendix A. 

Two questionnaires were developed to measure indicators of quality of care for residents who currently lived in the facility: one for the resident [1] and one for the most involved professional caregiver [2]. To be able to question every resident, we decided, in consultation with the experts, to create an adapted version of the questionnaire for residents who needed help to fill out a questionnaire because of physical or mental health issues. The questions in this version are the same as the questions in the standard resident questionnaire, but are reformulated from second to third person. They can be read to the resident or filled out by the resident’s informal caregiver (or professional caregivers if no informal caregiver was noted in the patient record), preferably together with the resident. 

A questionnaire was developed for the closest family caregiver (as noted in the health record) [3] and a separate one for the most involved professional caregiver [4] in order to measure indicators of quality of care for residents who passed away in the facility within the last six months. We performed a cognitive testing for all questionnaires in the corresponding responder group (i.e., residents, family, and professional caregivers). We tested the comprehensibility and response burden: recommendations resulted in minor linguistic changes for both residents and family caregivers. 

### 2.3. Step 2: Pilot Testing 

#### 2.3.1. Design 

We used a mixed-method design, including a quantitative application of the quality indicators and qualitative interviews with the nursing home staff using the instrument, in order to evaluate the face validity, feasibility, discriminative power, and usefulness of the instrument. 

#### 2.3.2. Setting and Participants

Nursing homes were recruited on a voluntary basis through a call for participation via involved community-based organizations. From the 24 candidates, we selected a purposive sample of nine nursing homes, while considering the number of beds (between 64 and 290 beds), the organizational structure (i.e., six profit and three non-profit) and the geographical location (every Flemish province was represented). 

Nursing homes were able to measure the quality indicators via questionnaires through a cross-sectional inclusion design. This method allows for nursing homes to gather information on residents who were currently living in the nursing home as well as those who had passed away. Following inclusion criteria were used:

Residents who were currently living in the nursing home and:lived for a minimum of one month in the facility;

Residents who had passed away and:lived for a minimum of one month in the facility; and,passed away four weeks to six months earlier in the nursing home.

#### 2.3.3. Measurement Procedure

All nursing homes followed the same measurement procedure based on a previously developed and tested method in order to measure the quality indicators via questionnaires [16]. Before the start of the pilot test, a coordinator per nursing home was appointed in consultation with the researchers. The researchers visited the coordinator (in the nursing homes) in order to explain the study, expectations, measurement procedure, how to work with the online questionnaires, going through the detailed instruction manual. The coordinator responsibilities include the supervision of the measurement procedure, communication within the nursing home (e.g., informing the staff about the instrument and procedure, announcing start date), drafting the list for including residents (in concordance with the researcher), and distribution of the questionnaire among residents, family caregivers and staff (Figure 2). The coordinator was also asked to keep a diary and note thoughts regarding the workload, setbacks, and/or other findings (Figure 2). 

Nursing homes were asked to include minimum of 2/3 of all residents at random and all deceased residents who met the inclusion criteria. We developed an inclusion matrix, depending on the number of residents per nursing home. We used online questionnaires via Limesurvey because nursing home staff were responsible for the distribution of questionnaires and to ensure responders privacy. No IP addresses were saved to guarantee anonymity. Residents could fill out the questionnaire via portable computers or tablets available in the nursing homes; family members received a link to the questionnaire via email; inhouse caregivers accessed the online questionnaire via computers in the nursing homes or on their private computers.

#### 2.3.4. Feedback and Evaluation

Per nursing home, a report was created, summarizing the individual and overall performance scores in a structured and standardized way. Nursing home coordinators were responsible for communicating the results to the nursing home staff (step 3 and 4 in Figure 2). After the report was sent to the nursing homes, the researchers visited the nursing homes for an evaluation interview with the coordinator, while using an interview guide with open-ended questions. The coordinators kept a diary during the measurement and delivered it in advance to the researcher. During the interview, the workload of the coordinator and the nursing home staff was evaluated and barriers and facilitators in the use of the quality indicators were identified. Coordinators could also share their thoughts on future use of the instrument and wider implementation. The researcher kept a diary of evaluation points, remarks, and questions for further qualitative evaluation during the whole period of the pilot test.

#### 2.3.5. Analyses

Data collection was closed after one month. Performance scores (non-adjusted mean) per quality indicator were calculated while using the defined numerators and denominators (range 0–100). In order to evaluate feasibility and discriminative power for individual quality indicator, we used descriptive and psychometric analyses in Microsoft Excel and SAS. Furthermore, the interviews were conducted with all coordinators in order to evaluate the face validity and usability of the indicators and the feasibility of the procedure. Together with the diaries of the coordinators and the field notes of the researcher, these interviews were analyzed while using a thematic framework approach, which was based on the barriers and facilitators for implementation framework of Grol and Wensing [10,39,40]. Table 1 presents an overview of all evaluation aspects, accompanying methods, and criterions.

### 2.4. Ethical and Language Issues

This study is approved by the Ethical Review Board of Brussels University Hospital of the Vrije Universiteit Brussel (protocol: QPACWZC01 BUN: 143201838240). All of the respondents (i.e., residents, family, and nursing home staff) received an online questionnaire, including cover letter and informed consent. Only questionnaires with signed informed consent were used to calculate performance scores. No IP addresses, names, or other personal identifiers were saved in the online questionnaire system. 

All of the indicators and questionnaires were developed and evaluated in Dutch. All of the interviews and trainings were performed in Dutch. The English translation was done specifically for this article. Dutch versions of the indicators or questionnaires are available on request.

## 3. Results

### 3.1. Step 1: Indicator Development 

#### 3.1.1. Phase 1 and 2: Literature Study and Expert Interviews

Based on the existing QPAC quality indicator set for specialized palliative care, the additional literature search (phase 1) and interviews with relevant stakeholders (n = 10) (phase 2), we identified 26 candidate quality indicators in eight domains of quality of palliative care for elderly persons in nursing homes. Table 2 shows the difference between the Q-PAC domains (specializes palliative care) and the domains for the nursing homes based on literature search and stakeholder interviews in phase 2.

#### 3.1.2. Phase 3: Expert Consensus 

Based on their individual evaluation of the 26 candidate quality indicators, seven quality indicators were immediately accepted and included. None were immediately rejected, so the remaining 19 quality indicators were debated in a one-day plenary discussion until consensus was found. Nine of 19 quality indicators were eventually accepted and three were newly developed during the meeting and added to the draft set. After the discussion, a set of in total 19 quality indicators were drafted and per email consented by all experts (Table 3 and Table 4). In Appendix A, the full list of quality indicators, as was tested in the pilot phase, is presented with accompanying numerator, denominator, question, and source.

### 3.2. Step 2: Pilot Test

#### 3.2.1. Responder Characteristics

Nine nursing homes tested the quality indicator set and measurement procedure. In total, 294 residents, 393 professional caregivers (345 for residents who currently lived in the facility and 48 for deceased residents), and 34 family caregivers completed the whole questionnaire and hence were included for the pilot study. We asked nursing homes to list the total number of inclusions, but four of them did not perform this assignment correctly; hence, we lack information on the response rates in this study. In total 214 of the residents were female and the majority (53%) of residents was between 85- and 94-years old. Table 5 presents an overview of characteristics.

#### 3.2.2. Psychometric Analyses: Feasibility and Discriminative Power 

None of the indicators had too many missing (>10%) answers. The quality indicators showed good discriminative power, as there were no indicators that had 95% or more answers in an extreme category (Table 3). Only two indicators had a variation range (min-max) smaller than 20 percentage points between different nursing homes, i.e., ‘Recognizing the approaching death’ and ‘Satisfied by care delivered’ and, hence, showed problems with sensitivity to change. 

#### 3.2.3. Qualitative Analyses; Feasibility, Usefulness and Face Validity

We interviewed all nine coordinators of the included nursing homes. With regard to face validity, all of the coordinators confirmed that the appeared to reflect their practice and seemed valid. As indicated by one on the coordinators: “*The results indicate clear work points and results are recognizable*”. They also agreed the results were easy to interpret and useful in terms of improving their service, but they indicated that they struggled in establishing concrete improvement goals that are based on the quality indicator scores. The coordinators evaluated the length of all four questionnaires as feasible, but four coordinators declared that they would prefer paper questionnaires for residents and family, as this may improve response rates. As indicated by one of the coordinators: “*We would prefer paper questionnaires … we* [staff in de nursing home] *don’t have professional email addresses and I didn’t want to send the questionnaires to their private email. Also, our residents don’t know how to use a computer or tablet and therefore some residents who normally could fill in a questionnaire alone, now couldn’t*”. 

Moreover, all of the coordinators indicated that they would use the instrument again and evaluated the instruction manual as useful and sufficient and assumed the instrument could be executed without the researchers. One of the coordinators said: “*Training was okay, the manual is clear and I think we could have managed without* [the manual]”. Additionally, all of the coordinators indicated that the workload was feasible and worthwhile, although they declared that preparing the list of respondents was time-consuming, as they had to acquire their own approach.

Based on field notes, the interviews with the coordinators, and diaries of the same coordinator, we made an overview of facilitators and barriers regarding the use of the instrument (Table 6) in general terms and per step of the measurement procedure, as described in Figure 2.

## 4. Discussion

In this study, we developed and evaluated a quality indicator set and a tailored measurement procedure consisting of 19 indicators to monitor the quality of palliative care in Belgian nursing homes. The composition of this indicator set is based on previously developed quality indicators for specialized palliative care, but, after adaptation to the nursing home context by experts and stakeholders, the themes differ somewhat: more emphasis is placed on autonomy and dignity of the nursing home residents. From this first pilot study, the quality indicators seem to be valid and the measurement procedure feasible for caregivers in nursing homes who are interested in improving the quality of end-of-life care within their center. From the psychometric analyses, we found that most of the quality indicators were feasible and they showed good discriminative power. The instrument appeared to reflect practice and hence confirmed face validity, according to coordinators during the qualitative interviews. The measurement procedure was evaluated by the interviewed coordinators as feasible and they indicated the measurement of the quality indicators could be performed based on the manual without extra help of the researchers. Overall, this study shows that the quality indicators are ready for further use in a large implementation study in Flemish nursing homes in order to further evaluate their feasibility, usefulness, discriminative power, and potential for quality improvement.

An evaluation of the quality of care with quality indicators best includes process as well as outcomes indicators of care in one monitoring cycle [16,41]. The quality indicator set for palliative care that we developed for nursing homes uses both types of indicators. We also included objective as well as subjective quality indicators. The psychometric analyses in this pilot study showed good results for all of the indicators on discriminative power expect for ’recognizing the approaching death’ and ‘satisfied by care delivered’, which are both indicators subjectively measured by caregivers. Both of the indicators might have been influenced by response bias, due to social desirability or a tendency to overestimate their skills [42] and were discarded from the quality indicator set. From this finding we might conclude that, when using self-assessment instruments for quality monitoring, caregivers should report as much as possible on objective information of care, i.e., information that can be found in the patient file. Such biases can best be monitored by regularly evaluating the quality indicator set for psychometric criteria, in order to keep the quality data sensitive to changes in quality of care over time and between health care services.

An important strength of our study is the rigorous, systematic development method while using stakeholders and the mixed-method design, including the RAND/UCLA method for indicator development, quantitative analysis of data, and qualitative interviews with the coordinators in the nursing homes to evaluate the instrument. Hence, we were able to evaluate the instrument and its measurement procedure in terms of face validity, feasibility, discriminative power, and usefulness. Additionally, because the thoroughly follow-up with the involved coordinators before, during and after the pilot, barriers and facilitators influencing the course of the measurement were identified. The small database is one of the limitations of this study. Psychometric analyses were limited and a study on further implementation is necessary to evaluate and validate the instrument including the quality indicators. The absence of response rates is another limitation of our study. Although coordinators drafted a list of included residents, we were unable to match them with the questionnaires because of GDPR policies. Additionally, although we aimed to include as many residents as possible while using three versions of the questionnaire in the resident’s evaluation, we have no insights regarding whether residents with cognitive problems, such as dementia, were sufficiently involved in the quality monitoring. 

Worldwide initiatives have been taken to monitor and improve the quality of palliative care in different settings [43,44,45,46,47]. Several studies have pointed out that the quality of dying and end of life care is not optimal across Western countries [48,49,50]. Some of these studies also used quality indicators in order to evaluate quality of palliative care in this setting, albeit being mostly focused on cancer patients, hospital and home setting, and administrative data in order to gather information on care processes and patient outcomes. Therefore, these measures are labeled as objective indicators and, although they provide a good basis for quality monitoring, they are not enough to point out strengths and weaknesses in specific long-term care organizations. Additionally, user perspective needs to be considered through subjective quality measures [51]. With our instrument, we focused on nursing homes and tried to combine both objective and subjective measures into different stakeholder perspectives in order to reach a comprehensive picture on quality of palliative care. Only this way, important themes for elderly persons, such as dignity and autonomy, can be properly addressed in order to improve the quality of care in the light of also improving the quality of life for residents in nursing homes. According to our qualitative analysis in this pilot study, coordinators indeed found the results of their measurement recognizable for their nursing home, supporting the face validity of these indicators (i.e., they are measuring what they aim to measure) from a caregiver point of view. This is an important finding, because, in order to reach effective change in health care, the value of timely and recognizable feedback is a crucial incentive for caregivers in order to continuously engage in these monitoring and improvement processes [11,40,45,52,53,54]. 

In light of care improvement in the field of palliative care in a nursing home context, a large-scale research project, ‘PACE steps to success’, has recently been implementing a combination of tailored improvement initiatives focusing on communication, advanced care planning, and knowledge and skills on end of life care while using a train-the-trainer implementation model. Although the intervention did not show significant effect on their primary outcome (comfort in the last week of life for residents), the process evaluation showed that the implementation rate was highly variable between countries and teams, and several challenges arose, such as attitude and motivation of staff, and skills and expertise of the trainer appointed to the individual nursing homes [26,48]. Our previous implementation research in palliative care already showed that caregivers are willing to invest in quality improvement trajectories and learn from other teams, but they need support from their management and financial reimbursement or staff to engage in these activities [40]. In this pilot study, we found the same barriers and facilitators pointing out the importance of setting the right preconditions for implementation in the nursing home context, throughout research and policy. This might be done by primordially evaluating nursing home readiness in order to increase the use and correct application of the quality indicators [55].

## 5. Conclusions

In this study, we developed and evaluated a quality indicator set and a tailored measurement procedure consisting of 19 indicators to monitor quality of palliative care in Belgian nursing homes. We combined both objective and subjective measures into four questionnaires for different perspectives in order to reach a comprehensive picture on quality of palliative care, end-of-life care and advance care planning in nursing homes. Care teams in nursing homes are able to monitor themselves based on these indicator scores. We found, while using both quantitative as qualitative analyses, the developed instrument had good face validity, feasibility, discriminative power, and it is useful in terms of quality monitoring according to caregivers, though establishing concrete improvement goals based on quality indicator scores remains difficult for them. The quality indicators are ready for further use in a large implementation study and process evaluation in Flemish nursing homes in order to further evaluate their feasibility, usefulness, discriminative power, and potential for quality improvement.

## Figures and Tables

**Figure 1 ijerph-18-00829-f001:**
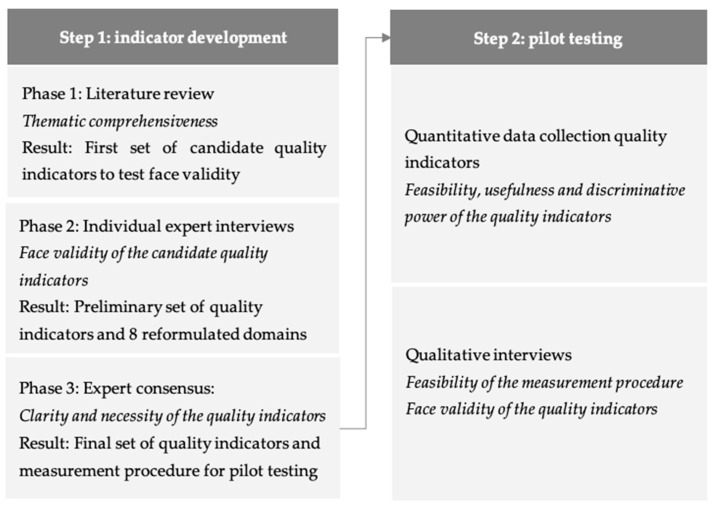
Overview of study phases and accompanying results.

**Figure 2 ijerph-18-00829-f002:**
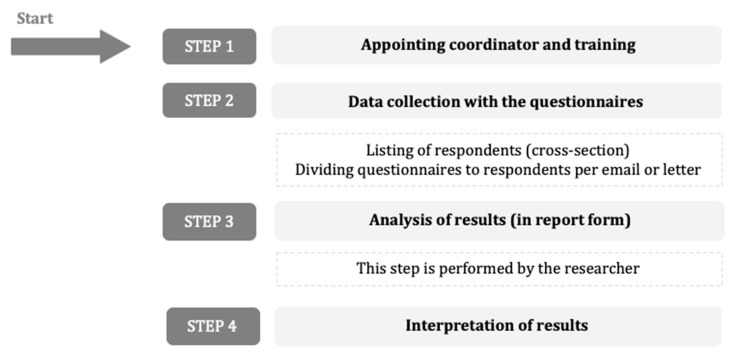
Measurement procedure.

**Table 1 ijerph-18-00829-t001:** Overview of the evaluation and accompanying methods and criterions.

Aspect	Definition	Evaluation Method	Criterion to Judge Aspect as Adequate
*Individual quality indicators (QI’s)*			
Face validity	The extent to which QI’s are subjectively viewed as covering the concept it purports to measure	Qualitative: interview: feedback on every single quality indicator was asked in terms of face validity	Subjective confirmation of validity of quality indicator scores
Feasibility	The extent to which the QI’s are measurable	Quantitative: psychometric analyses	Not more than 10% missing values per question
Discriminative power	The extent to which a QI discriminates between good and bad quality	Quantitative: psychometric analyses	Not more that 95% of answers in an extreme category Meaningful range between QI scores (min–max ≥20%)
Usefulness	The extent to which the QI scores can be used to improve care	Qualitative: interview question “Were you able to define improvement point based on the quality indicator scores and feedback report?”	Subjective confirmation of usefulness
*Overall quality indicator measurement*			
Feasibility	The extent to which the measurement procedure is feasible for caregivers in nursing homes	Qualitative: interview question “Do you have the feeling you are able to measure the quality indicators without any support in the future?”	Subjective information on work-load for caregivers
		Qualitative: interview question “How did you feel about the length of the questionnaire?”	Subjective information on survey completion time for caregivers

**Table 2 ijerph-18-00829-t002:** Eight thematic domains for quality indicators.

Original QPAC Set [24]		QPAC for Nursing Homes	
1	Physical aspects of care	Physical aspects of care	1
2	Psychological, social and spiritual aspects of care	Psychological, social and spiritual aspects of care	2
3	Care planning, information and communication with patients	Autonomy and dignity	3
		Care planning and communication with residents	4
4	Care planning, information and communication with family	Communication with family	5
5	Care planning, information and communication between caregivers	Communication between caregivers	6
6	Circumstances surrounding death	Care and circumstances surrounding death	7
7	Coordination and continuity of care		
8	Support for family	Care for family	8

**Table 3 ijerph-18-00829-t003:** Participants in expert consultation rounds.

	Total
**Professional caregivers from care homes**	**7**
Head nurse/Referent nurse	3
Paramedic	1
Care personnel	1
Physician	1
Quality coordinator	1
**Representatives from residents and next-of-kin**	**3**
Flemish Expertise Centre for Dementia	1
Alzheimer League, family council	1
Flemish elderly council	1
**Palliative care research and policy**	**5**
KU Leuven—LUCAS research group	2
Flemish Federation Palliative Care	1
Local Palliative home care network Westhoek-Oostende	1
Flemish agency for care and health	1

**Table 4 ijerph-18-00829-t004:** Quality indicators for palliative care in nursing homes.

**Domain: Physical Aspects of Care**					
N	Short title	Description of the indicator	Respondent	Mean score (%)	Range (min–max)
PC-1	Being in pain	Percentage of residents with a pain score of 3 or more in the last three days	Residents	30.7	37.1 (19.1–56.3)
**Domain: Psychological, Social and Spiritual Aspects of Care**					
N	Short title	Description of the indicator	Respondent	Mean score (%)	Range (min–max)
PC-2	Feeling worried or anxious, or a burden	Percentage of residents who indicate they were most of the times or always feeling worried or anxious, or a burden to others	Residents	9.2	23 (4.3–27.3)
PC-3	Being around people who care about you	Percentage of residents who indicate that they were most of the times or always able to be around people who cared about them	Residents	57.1	42.9 (29.8–72.7)
**Domain: Autonomy and Dignity**					
N	Short title	Description of the indicator	Respondent	Mean score (%)	Range (min–max)
PC-4	Personal wishes and beliefs respected	Percentage of residents who indicate that their caregivers most of the times or always respecting their personal wishes and beliefs	Residents	63.3	55.7 (35.2–90.9)
PC-5	Decisions about life and care	Percentage of residents who indicate that they most of the times or always can make their own decisions about their life and care	Residents	44.2	35.4 (31.3–66.7)
PC-6	Treated with respect	Percentage of residents who indicate that they most of the times or always were treated with respect	Residents	68.6	47.2 (43.8–90.9)
**Domain: Care Planning and Communication with Residents**					
N	Short title	Description of the indicator	Respondent	Mean score (%)	Range (min–max)
ACP-1	Information comprehensible and not contradictory	Percentage of residents who indicate that they most of the times or always receive comprehensible information and almost never of never contradictory information	Residents	79.5	21.4 (72.3–93.8)
ACP-2	Conversation with family	Percentage of residents for whom the next-of-kin indicates that more than once a conversation took place with the caregivers, the next-of-kin and, when possible, the resident	Next-of-kin	47.6	100 (0–100)
ACP-3	Knowledge about care goals and life wishes	Percentage of residents for whom their professional caregiver indicates that they have knowledge about the residents’ care goals and life wishes.	Professional caregiver	63.8	40 (47.1–87.1)
ACP-4	Encouraging ACP	Percentage of residents for whom their professional caregiver indicates that they often or very often encourage residents and their next-of-kins to involve in advance care planning.	Professional caregiver	37.7	72.5 (10.8–83.3)
**Domain: Communication with Family**					
N	Short title	Description of the indicator	Respondent	Mean score (%)	Range (min–max)
ACP-5	Next-of-kin involved in decisions	Percentage of next-of-kin who indicate that they often or very often felt involved in the decisions taken about the resident.	Next-of-kin	64.7	75 (25–100)
EOL-1	Information about approaching death	Percentage of next-of-kin who indicate that they received the right amount of information on the approaching death of the resident.	Next-of-kin	73.5	35.7 (64.3–100)
**Domain: Communication between Caregivers**					
N	Short title	Description of the indicator	Respondent	Mean score (%)	Range (min–max)
PC-8	Information in resident file	Percentage of residents for whom the professional caregiver finds sufficient information in the resident file when needed.	Professional caregiver	69.3	37.6 (52.7–90.3)
**Domain: Care and Circumstances Surrounding Death**					
N	Short title	Description of the indicator	Respondent	Mean score (%)	Range (min–max)
EOL-3	Comfortable in last week of life	Percentage of next-of-kin who indicate that many or a lot of measures were taken to make the resident comfortable in the last week of life.	Next-of-kin	67.6	100 (0–100)
EOL-4	Recognizing the approaching death	Percentage of residents for whom the professional caregiver indicates they could recognize the approaching death well or very well by physical changes.	Professional caregiver	91.7	16.7 (83.3–100)
EOL-5	Satisfied by care delivered	Percentage of residents for whom the professional caregiver indicates they are satisfied with the care delivered to the resident.	Professional caregiver	95.8	16.7 (83.3–100)
EOL-6	Support by specialized palliative care	Percentage of residents for whom the professional caregiver indicates a palliative care referent or specialized team was involved in the care for the resident.	Professional caregiver	68.8	100 (0–100)
		**Domain: Care for Family**			
PC-7	Attention for wishes and feelings of next-of-kin	Percentage of next-of-kin who indicate that the professional caregivers had attention for their wishes and feelings.	Next-of-kin	67.6	30 (50–80)
EOL-2	Supported immediate after death	Percentage of next-of-kin who indicate that they felt sufficiently supported by the professional caregivers immediate after the death of the resident.	Next-of-kin	85.3	66.7 (33.3–100)

PC = Palliative care; ACP = advance care planning; EOL = end of life.

**Table 5 ijerph-18-00829-t005:** Characteristics per response type in the pilot test.

Response Type	Total	Female (%)	Age of Resident				Dementia ^B^ (%)	Length of Stay ^C^		
			<75 (%)	75–84 (%)	85–94 (%)	>94 (%)		<12 (%)	12–24 (%)	>24 (%)
**Residents**	**294**	**214 (73)**	26 (9)	74 (25)	157 (53)	37 (13)	NA	NA	NA	NA
*Resident him/herself*	114	83 (73)	11 (10)	24 (21)	67 (59)	12 (11)	NA	NA	NA	NA
*Together with family caregiver*	63	43 (68)	5 (8)	14 (22)	38 (60)	6 (10)	NA	NA	NA	NA
*Family caregivers in the name of the resident*	116 ^A^	87 (75)	10 (9)	35 (30)	52 (45)	19 (16)	56 (48)	37 (33)	22 (20)	52 (47)
**Professional caregivers**	393	305 (73)	27 (7)	97 (25)	218 (55)	51 (13)	204 (49)	125 (32)	60 (15)	208 (53)
*Residents who lived in the facility*	345	257 (74)	25 (7)	88 (26)	193 (56)	39 (11)	162 (47)	109 (32)	54 (16)	182 (53)
*Deceased residents*	48	31 (65)	2 (4)	9 (19)	25 (52)	12 (25)	29 (60)	16 (33)	6 (13)	26 (54)
**Family caregivers**	34	22 (65)	1 (3)	7 (21)	17 (50)	9 (26)	16 (47)	14 (41)	5 (15)	15 (44)

A: 5 missings for length of stay. B: Questioned only when the family caregivers completed the questionnaire in the name of the resident. C: Length of stay in months.

**Table 6 ijerph-18-00829-t006:** Facilitators and barriers based on the interviews and dairies regarding the use of the instrument.

Barrier (b) or Facilitator (f)	Quote from Caregivers or Field Notes	Diary by Coordinator	Interview with Coordinator
The use of the instrument in general terms			
Lack of time and staff to perform quality measurement (b)	“*To sell the instrument: make it a sort of an obligation, otherwise it will not happen, I think. So much extra is added [next to the regular work], and also many projects that are already there anyway*” (coordinator nursing home)	X	X
Readiness of the team to perform quality monitoring together (f)	“*[experience with implementation of the quality assessment] it was ok. It also depends on the enthusiasm and commitment of the persons who are doing it.*” (coordinator nursing home)		X
Step 1: Appointing coordinator			
Presence of a good coordinator to guide the quality measurement (f)	*“Appointment of the coordinator: one is not enough. Depends on the size of the nursing home.”* (coordinator nursing home) *“Announced [the quality assessment] during team meeting. They [coordinators] had made a step-by-step plan and mailed it to the staff, how they could easily find it and fill it in … everything went smoothly”* (coordinator nursing home)	X	X
Step 2: Data collection with the quality indicators			
Bad timing regarding the start of measurement (i.e., sick staff, loss of coordinator) (b)	Some of the coordinators became absent during the procedure and the person who took over didn’t have all the needed paperwork. (field notes researchers). Some nursing homes forgot to record the total of included participants, didn’t sent out the recruited number of questionnaires or didn’t sent questionnaires to family caregivers. The reason they indicated was the moment of the measurement was not convenient (field notes researchers).		X
Lack of computer literacy in all participants (b)	“*They [family and residents] had no e-mail and some [family] had to come to the nursing home to fill it [the questionnaire] in.*” In some nursing homes professional caregivers didn’t had a work email and in one of these homes, the coordinator had to aid each included professional caregivers with opening the link [which made available on the desktop] to the questionnaire (field notes researchers).		X
Lack of technology in the nursing homes (b)	*“It was a lot of time investment, there was only one iPad available in the nursing home, so we had to arrange a lot.* *WIFI connection was also not reliable, which limited usability.”* (coordinator nursing home)	X	X
Feasible workload (f)	All coordinators found the overall workload feasible (field notes researchers) *“A lot of work in preparation by the coordinator so the coordinator should certainly have time to prepare. Once it runs [there is] little follow-up work.”* (coordinator nursing home)	X	X
Step 3: Analysis of results by researchers			
Low(er) response rate because of measurement procedure (b) AND Inclusion of deceased residents due to low mortality (b)	*“With a longer measurement period, they [respondents] could fill in more”* (coordinator nursing home)	X	X
Fast (within two weeks) analysis of questionnaires because of the use of digital data (f)	Because we used online questionnaires the researchers didn’t need to input any data but could directly analyse resulting in fast feed-back to the nursing homes		X
Step 4: Interpretation of results by coordinator and nursing home team			
Easy to interpret results (f)	“*The results indicate clear work points. Results are recognizable*” (coordinator nursing home)		X
Struggle to go from interpretation to establishing improvement goals (b)	Most coordinators indicate they recognize the results, but they cannot (yet) make clear improvement goals. (field notes researchers)		X

## Data Availability

The data presented in this study are available on request from the corresponding author.

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
