# Peer review of "Pilot Study to Develop and Test Palliative Care Quality Indicators for Nursing Homes"

_ijerph, 2021, doi:10.3390/ijerph18020829_

Round 1
Reviewer 1 Report
In my opinion, the authors present a good study for measuring the quality of palliative care for nursing homes. The methods were described in detail and the authors developed the indicators thoroughly which can make a good criteria for assessing the palliative care quality.
I have only two suggestions:
in line 94-98, the authors should describe more specific of where the literature is searched (pubmed? Embase?) or what existing indicators are.
In line 244, “Domain: Psychological, social or spiritual aspects of care” should be in gray background.
Reviewer 2 Report
- Please summarize the process and results of phase 1 and phase 2 briefly in the rection of results ( line 229 ).
- Please check the typing of headings in Table 2 , Table 3 , and the Domain in Table 4.
- Some of the thematic domains between Table 2 and Table 4 are different. The reasons have to be described in the text.
- The overview of the characteristics of the residents in Table 5 have to be described briefly.
- The themes of this research reinforced autonomy and dignity of the nursing home residents ( lines 293-294 ) , it needs to be discussed further.
Reviewer 3 Report
Some sections of the manuscript are difficult to follow. It may be useful to present tools and results in a more concise way.
The search for a tool on palliative care is fundamental to improve the quality of care and life expectancy. The work is well structured, however a more schematic presentation of the research and results would be needed.
